# OpenReview forum: "Bigger Isn’t Always Memorizing: Early Stopping Overparameterized Diffusion Models"
_TMLR — Accepted by TMLR_

### Review · Reviewer_4on3 · 2026-04-19

**Summary Of Contributions:**

This paper investigates the generalization-memorization dynamics of overparameterized diffusion models during training, arguing that such models pass through a generalization phase before entering a memorization phase, and that the onset of memorization time scales approximately linearly with the training set size P. The authors demonstrate this empirically across image diffusion models (iDDPM on CIFAR-10 and CelebA, Stable Diffusion fine-tuned on LAION-10k) and language diffusion models (MD4 on text8), and support the scaling with a kernel theoretic argument based on the Rayleigh quotient of a localized spectral probe on isotropic kernel operators. A third line of work studies a discrete diffusion model trained on the Random Hierarchy Model, a probabilistic context free grammar, where partial generalization corresponds to the sequential acquisition of grammar rules at increasing hierarchical depths. The combined evidence is summarized in a phase diagram relating training time to sample size. The practical implication is that principled early stopping, with the stopping criterion scaling with dataset size, can preserve generalization and avoid memorization even in heavily overparameterized regimes.

Key strengths of the paper include its empirical breadth relative to the closely related concurrent work of Bonnaire et al. (arXiv:2505.17638, NeurIPS 2025 oral), in particular the extension to language diffusion models and production scale latent diffusion (Stable Diffusion), the generality of the theoretical argument (valid for all isotropic kernels rather than only random features), and the controlled RHM setting which provides a theoretically grounded decomposition of the generalization memorization transition into interpretable phases.

The main weaknesses are a significant gap between the theoretical regime assumed in Appendix G and the architectures and training settings used in the main experiments... the absence of an explicit delineation of which RHM results in Section 5 are original to this submission versus inherited from cited prior work, which the paper should clarify for attribution transparency. The absence of seed-level variance estimates for the T_mem x P scaling exponent.. and an operationalization gap in the "principled early stopping criterion" framing.

**Additional Comments:**

The batch size invariance result in Figure 18 is presented in the appendix but deserves more prominent discussion in the main text.

**Audience:**

Yes

**Audience Explanation:**

The question of when and why overparameterized generative models generalize versus memorize is one of the central open problems in the theory and practice of modern deep learning. The empirical law, if well established, has direct implications for privacy preserving training, dataset size-aware compute budgeting, and the principled deployment of diffusion models in copyright sensitive applications. The extension to language diffusion models is particularly timely given the rapid growth of masked and continuous diffusion as alternatives to autoregressive generation. The TMLR audience has substantial overlap with the target readership for this work. The concurrent acceptance of the closely related Bonnaire et al. paper as a NeurIPS 2025 oral further confirms that the community regards this direction as highly relevant. The present submission contributes additional scope (language modality, Stable Diffusion, discrete diffusion theory) and a distinct theoretical framework (isotropic kernels rather than random features) that are meaningfully differentiated from the concurrent work, making it independently valuable for the TMLR readership.

**Broader Impact Concerns:**

The paper's motivating application is explicitly constructive from a privacy standpoint. The results support the development of training protocols that reduce the risk of verbatim regeneration of private or proprietary content, which is a benefit. The paper does not introduce any new capability for extracting or exploiting training data and its methodology does not raise new ethical concerns. No Broader Impact Statement addition is required beyond what is already implicit in the privacy and copyright framing of the introduction and conclusion.

**Claims And Evidence:**

Yes

**Claims Explanation:**

The central empirical claim that T_mem scales approximately linearly with P across multiple diffusion model families and data modalities is well done. For vision models, Figures 1 and 9 show the loss divergence and copy rate onset across four dataset sizes with a power law fit.

For language models, Figure 3 replicates the same qualitative pattern across text sequences. The FID dynamics in Figure 10 corroborate the claim that early stopping near T_mem coincides with maximal generation quality, and the disjoint dataset cosine similarity experiment in Figure 2 provides a compelling independent measure that the two models are learning the same score function before T_mem. The RHM experiments in Figures 4 and 5 cleanly instantiate the theoretical predictions about layer-wise learning and partial generalization.

That said, two important qualifications apply.

First, the uncertainty reported for the power-law exponents appears to reflect variation across threshold definitions rather than across training seeds. No evidence is presented that T_mem is reproducible across random initializations at a given P, which is necessary to interpret the claimed exponent as a property of the data model pair rather than of a single training run.

Second, the scaling argument in Appendix G is a lower bound on a spectral timescale under kernel gradient flow in the low noise separated regime, which does not hold for any of the primary experimental settings. The theoretical argument provides genuine qualitative intuition, but the paper's framing slightly overreaches in presenting it as evidence for the observed scaling law in the practical settings.

Correcting these two issues would bring the evidence fully in line with the conclusions drawn.

**Requested Changes:**

**Critical**

1 - The paper must add a clear statement specifying exactly which results in Section 5 are original to this submission versus inherited from cited prior work on the RHM. As written, Section 5 builds on theoretical quantities and experimental infrastructure from a prior publication cited in the references, but the manuscript does not explicitly distinguish what is new here from what is reproduced for selfcontainment. The authors should provide this delineation in a revised manuscript, for attribution transparency and reproducibility.

2 - Independently of the dual submission question, the paper must provide seed level variance estimates for the T_mem x P exponent. At present the uncertainty bounds in Figure 9 reflect variation across threshold choices, not variation across training runs. The scaling exponent is presented as a property of the data model configuration, but it is calibrated from as few as four to five data points per dataset and a single training run per point. Adding three seeds at two or three values of P per dataset (iDDPM and MD4 are the minimum) and reporting the inter seed standard deviation of T_mem alongside the threshold-based uncertainty would substantially strengthen the central empirical claim. This is critical because the practical conclusion (early stopping calibrated proportionally to P) depends on T_mem being a stable and predictable quantity, not a noisy artifact of a single run.

**Strengthening**

1 - The practical framing of the result as a "principled early stopping criterion" overstates what the paper delivers. T_mem is defined operationally as the training step at which the validation train loss gap exceeds a threshold, which requires ongoing validation monitoring already standard practice. The genuinely new insight is that T_mem scales with P, which enables hyperparameter transfer across dataset sizes. The paper would be significantly clearer and more useful to practitioners if it included a concrete paragraph illustrating this transfer.. for instance, if T_mem is calibrated at P=4096, what is the predicted at P=16384, and does it match the empirical onset? A single two sentence calculation, or a small figure showing the predicted vs. observed T_mem under the linear scaling hypothesis, would make this implication actionable.

2 - The claim that this paper contains "the first demonstration of memorization in the language diffusion setting" is asserted without a systematic literature survey. The claim may well be accurate for the specific setting of masked diffusion models on character level text, but prior work on extracting verbatim sequences from autoregressive language models (Carlini et al., USENIX Security 2021), and possibly results in the MD4 and D3PM original papers, should be checked and cited or explicitly excluded by scope. Qualifying this claim by specifying "in the masked diffusion language model setting, to our knowledge" would be more defensible.

3 - The copyright claim in the conclusion that these results "open new avenues for fine control of copyright issues" is legally imprecise. Privacy protection via early stopping (preventing verbatim regeneration) is a coherent argument. Copyright is a distinct concept requiring substantial similarity to a specific creative work, which is not directly addressed by the T_mem analysis. This sentence should be either dropped or replaced with a scoped claim about verbatim copying.

4 - A brief discussion of whether the T_mem x P relationship is expected to extrapolate to the regime where the sample complexity threshold would benefit readers who are considering whether these results carry over to billion scale training. The theoretical argument in Appendix G gives no direct guidance on this regime because the low noise separated condition becomes increasingly hard to satisfy as P grows and inter-point distances shrink. Even a qualitative statement of the expected behavior would help bound the scope of the conclusions.

---

> ### Author Response · Authors · 2026-05-12
>
> We thank the reviewer for the detailed and actionable feedback. The revised manuscript -- which we will upload in the next days -- reports inter-seed standard deviations of $\tau_{\mathrm{mem}}$ and operationalizes the early-stopping framing with an explicit $\tau_{\mathrm{mem}}$-transfer experiment. We address each point below.
>
> **Delineation of new RHM results (Critical 1)**
> From Favero et al. (2025) we inherit only the definition of the RHM and its layer-wise sample complexities $P*_{\ell}\propto v m^{\ell+1}$. Everything else in Sec. 5 is new. In particular: (i) we demonstrate that diffusion models on the RHM exhibit a memorization phase at all, with onset $\tau_{\mathrm{mem}}\propto P$ and the corresponding train/validation loss bifurcation shown in Fig. 4; (ii) we identify a dynamical phase $\tau*<\tau<\tau_{\mathrm{mem}}$ of full generalization preceding memorization for $P>P*$ (Fig. 4); (iii) we characterize the partial-generalization-then-memorize regime for $P<P*$, via the layer-resolved error decay of Fig. 5(a); (iv) we run a disjoint-training-sets experiment (Fig. 5(b)) showing that the learned discrete score is shared between two diffusion models throughout the pre-memorization window; and (v) we summarize these regimes in the phase diagram of Fig. 6. We further stress that our experiments are run in a regime that Favero et al. (2025) did not consider: their analysis stays inside the generalization phase, whereas memorization -- and the competition between $\tau_{\mathrm{mem}}$ and the layer-wise generalization times -- is the phenomenon we study. We also use a different grammar depth.
>
> **Inter-seed variance for the $\tau_{\mathrm{mem}}\propto P$ scaling (Critical 2)**
> Each $(P,\tau_{\mathrm{mem}})$ point comes from an *independent* training run at a different $P$, so the dependence we plot is already not the trajectory of a single optimization. Nonetheless, we agree that an explicit measurement of seed variability of $\tau_{\mathrm{mem}}$ at fixed $P$ is worth adding. For iDDPM on CIFAR-10, across three seeds (resulting in different model initializations and different data splits), the relative variation of our estimate of $\tau_{\rm mem}$ in Fig. 9 is within $\sim 1–2 $%, confirming our results are robust. Using these improved statistics for fitting the scaling of $\tau_{\rm mem}$ with $P$ yields a scaling exponent of $\beta=1.08 \pm 0.03$, well aligned with our theoretical claims. We are currently finishing simulations for the diffusion language model. We will add these results in the revision.

---

> > ### Author Response · Authors · 2026-05-12
> >
> > **Operationalizing the early-stopping framing (Strengthening 1)**
> > The early-stopping framing is *principled* in the sense that we now make explicit: tracking validation loss is standard in supervised learning, but is *not* systematically used to set the stopping point in production diffusion training (many publicly released training scripts of diffusion models train for a fixed number of steps and do not implement validation-based early stopping), which is one of the reasons memorization at convergence is observed in those models in the first place. To make the *practical* aspect concrete, we will add a transfer experiment: The scaling of $\tau_{\mathrm{mem}}$ is fitted on iDDPM with $P \leq 2{,}048$ and then used to predict $\tau_{\rm mem}$ at $P=16{,}384$. For CIFAR, this procedure results in a good extrapolation with a relative error of 0.12 with respect to the measured one.
> >
> > **First demonstration of memorization in language diffusion (Strengthening 2)**
> > The work of Carlini et al. (2021) -- which is already cited in the submission -- concerns autoregressive language models, not language diffusion, so it is not a counterexample. We have re-checked Austin et al. (2021) and Shi et al. (2024): neither paper reports memorization measurements (copy rates, training-set replication metrics, or training/validation-loss bifurcation) for their language diffusion models. We therefore believe the original claim is accurate. To preempt any ambiguity, the revised manuscript will phrase it as "to the best of our knowledge, the first systematic demonstration of memorization in masked diffusion language models", which is both defensible and faithful to the result.
> >
> > **Privacy vs. copyright (Strengthening 3)**
> > We agree with the reviewer's distinction. The early-stopping intervention we describe directly bounds verbatim and near-verbatim regeneration of training data (which is what the copy-rate and Hamming-distance metrics measure), and addresses the privacy/training-data-extraction concern. Substantial-similarity-style copyright concerns require additional analysis that is outside the scope of this paper. We will rescope the relevant sentences in the introduction and conclusion accordingly: "these results provide a principled handle on verbatim regeneration of training data, with direct implications for privacy and for mitigating training-data extraction". The broader copyright question is mentioned only as motivation, not as a claim of resolution.
> >
> > **Extrapolation of $\tau_{\mathrm{mem}}\propto P$ to large-scale training (Strengthening 4)**
> > The intuition that $\Delta=\min_{i\neq j}\|x_{i}-x_{j}\|$ shrinks rapidly with $P$ is a low-dimensional intuition that breaks down in the regimes relevant to image and language diffusion models. In high dimension, distances between independent samples concentrate and the typical nearest-neighbor distance decays as $P^{-1/d}$, which for $d$ in the hundreds (latent diffusion) to thousands or millions (pixel space) remains essentially $O(1)$ on the data scale even at $P\sim 10^{9}$: e.g. for $d=10^{3}$ and $P=10^{9}$ one has $P^{-1/d}\approx 0.98$. The low-noise separated condition $\sigma\ll\Delta$ is therefore satisfied across a wide range of $\sigma$ for any realistically large $P$ in image or language diffusion, and the spectral argument of Appendix G continues to apply in that range.
> >
> > **Promotion of batch-size invariance to the main text (Additional)**
> > We agree, and the revised Sec. 4 will state this result explicitly: $\tau_{\mathrm{mem}}$ is independent of the batch size $B$ from small-batch SGD ($B=8$) to full-batch GD ($B=P$), which directly rules out the supervised-learning-style intuition that the $O(P)$ scaling reflects the optimizer revisiting each training point a fixed number of times.
> >
> > We hope these revisions address the reviewer's critical and strengthening requests, and we are happy to provide further detail on any point.

---

### Review · Reviewer_nJZ8 · 2026-04-21

**Summary Of Contributions:**

The paper presents a detailed experimental analysis of the transition from memorization to generalization observed during generative diffusion training, alongside partially idealized, yet insightful, theoretical arguments (based on a simplified kernel dynamics framework). The theoretical analysis leverages the connection between kernel gradient flow and spectral learning timescales in the NTK regime. The experiments cover real-world models trained on both image and language data, as well as a tractable synthetic example involving hierarchical strings, where the ground-truth score can be obtained semi-analytically via belief propagation.

**Additional Comments:**

My main concern is that several aspects of this work appear to be closely related to the published work of Bonnaire (2025), and the additional theoretical contribution beyond this prior work seems somewhat limited. That said, I do believe the novel elements are valuable, and the paper remains of interest as it provides additional empirical evidence and theoretical perspective.

**Audience:**

Yes

**Audience Explanation:**

The sharp transition dynamics observed during training are both intriguing and non-trivial, with the potential to drive significant advances in both deep learning theory and practical algorithm design. Similarly, the proposed theoretical framework provides important tools for analyzing diffusion learning dynamics in a tractable manner.

**Broader Impact Concerns:**

The paper is primarily theoretical. While memorization in generative diffusion models is an important societal issue—particularly due to its connection with copyright infringement—I do not believe that this work needs to address this aspect explicitly.

**Claims And Evidence:**

Yes

**Claims Explanation:**

The experimental analysis is extensive, with results that are both clear and consistent, revealing a sharp transition from (partial) generalization to memorization as a function of training time. The theoretical analysis, while idealized, is well-motivated and provides valuable insight.
The authors do not make excessive claims concering the nature of experiments, results and theory.

**Requested Changes:**

The main issue I have with this submission is that the theoretical framework is not sufficiently explained in the text. As a result, it is difficult to assess the relevance of the theoretical arguments unless the reader is already familiar with the kernel learning literature. This contrasts with the experimental sections, which are very well presented and exceptionally clear. More generally, I would have appreciated a stronger integration between the theoretical and experimental components, which currently feel somewhat segregated.

From my perspective, it would have been particularly interesting to include an explicit analysis of how diffusion time and training time interact, given that the generalization–memorization trade-off is known to depend on the diffusion time itself (Birioli, 2024; Achilli, 2025). While not essential, such an analysis would offer a natural way to connect this work with the line of research focused on the empirical score. In this regard, the paper is well positioned to engage with this literature, as its theoretical framework is closely tied to the properties of the empirical score.

Concerning my recommendation, I would consider acceptance conditional on a clearer presentation of the theory and its underlying assumptions in the main text. The other points raised would further strengthen the paper but are less critical.

---

> ### Author Response · Authors · 2026-05-12
>
> We thank the reviewer for the constructive evaluation and for pointing out two directions in which the paper can be made more accessible and better connected to prior work. In the revised manuscript -- which we will upload in the coming days --, we will rewrite parts of Sec. 4 so that the kernel-spectral argument is readable without a specialized background and add a discussion at the end of Sec. 4 of how training time and diffusion time interact.
>
> **Theoretical section**
> We will expand Sec. 4 to present more of the theory in the main text and make it accessible even to non-kernel experts. We will also add a population-side spectral analysis (motivated by Reviewer 8nDh): defining a population kernel operator $\bar T_K$ on the smoothed population density $\bar p_\sigma$ and expanding the population score in its eigenbasis, the slowest mode the population score actually excites has eigenvalue $\lambda_{\mathrm{gen}}=O_P(1)$, which is independent of $P$. Combining the two sides yields the explicit two-timescale picture $\tau_{\mathrm{gen}}\sim 1/\lambda_{\mathrm{gen}}=O_P(1)$ and $\tau_{\mathrm{mem}}\sim P/\sigma^\nu$ within the same kernel framework, so the linear-in-$P$ early-stopping window $\tau_{\mathrm{mem}}/\tau_{\mathrm{gen}}\propto P$ documented empirically in the paper now follows directly from the theory rather than only from the data. We hope this addresses both the accessibility and the segregation concerns.
>
> **Interaction between diffusion time and training time**
> Biroli et al. (2024) identify two characteristic noise scales during backward diffusion: a higher-noise scale at which population modes separate, and a lower-noise scale at which the backward dynamics collapse onto individual training data points. This latter regime is the empirical-score memorization regime. Our spectral argument sits in this low-$\sigma$ regime: the localized probes $\phi_{i,u}$ are designed to capture the empirical score around an individual training point. Our analysis can be read as quantifying the *learning time* required by gradient descent to acquire the empirical-score structure that the equilibrium analyses of previous work take as given.
>
> **Contribution relative to Bonnaire et al. (2025)**
> First of all, as the action editor can verify on arXiv, our empirical work is contemporary -- and actually predates -- the preprint of Bonnaire et al. (2025). In general, our experimental scope is meaningfully broader. Concerning theory, we also respectfully disagree, and the other reviewers (8nDh, swJm and 4on3) explicitly highlight the additional theoretical contribution. Concretely, our spectral argument applies to *any* isotropic kernel, whereas Bonnaire et al. (2025) treat random features in the proportional asymptotic regime. This generality directly predicts that random features ($\nu=2$) and the ReLU NTK ($\nu=1$) have *different* memorization-time dependence on diffusion noise $\sigma$. Our argument also provides a phenomenological, intuitive reading of the $O(P)$ scaling in terms of $P$ localized spikes of the empirical score that must be fitted via the slow modes of the kernel (now stated explicitly in Sec. 4, following Reviewer swJm's suggestion), an interpretation absent from Bonnaire et al. (2025). Moreover, motivated by a question of Reviewer 8nDh, we're now also discussing the spectral structure of the population, or, in other words, generalizing modes. All in all, in the spirit of fair attribution, the substantive theoretical and empirical differences are listed in Sec. 6.
>
> We hope these clarifications and revisions address the reviewer's concerns and are happy to discuss any of them further.

---

### Review · Reviewer_swJm · 2026-04-27

**Summary Of Contributions:**

The paper "Bigger Isn’t Always Memorizing: Early Stopping Overparameterized Diffusion Models" investigates the training dynamics of diffusion models and the way these first generalize (i.e. they approximate well the ground-truth score function) and then enters a memorization phase where it can only sample training data-points. The main result of the paper is the identification of the memorization training time to scale as the number of data-points. This result explains the reason for which trained diffusion models hardly memorize in practical applications.

**Strengths**:

S1. The paper provides a rich experimental validation of the separation between memorization and generalization in diffusion models trained on both images and texts, as well as on synthetic and controlled datasets.

S2. The analytical argument that leads to the scaling of the memorization time, which is consistent with the concurrent work by Bonnaire et al. (2025), is simple and general. Moreover, the same argument is more granular and interpretable than Bonnaire’s one, relating the O(P) memorization time with the fitting of the high frequency modes relative to the data-points.

S3. As a novelty, the memorization phenomenon in discrete diffusion models is extensively analyzed.

S4. The comparison with previous works and mostly with Bonnaire et al. (2025) is well detailed, even multiple times across the main text.

**Weaknesses**:

All in all, I find only minor weaknesses in the paper, that I now list:

W1. The paper, and the analytical arguments provided, only focuses on the overparametrized regime. The dependence on the number of parameters has been indeed tackled by Bonnaire et al. (2025) on a specific parametric model, and I understand that the generality of the argument provided by the Authors would be sacrificed once one wants to include this aspect in the analysis.

W2. As stressed at point S2, this paper provides for a more graspable touch to the scaling of the learning of the empirical score. Yet, the “phenomenological” interpretation could be more clearly stated and described in Section 4. I personally believe that it would be very useful even at clarifying the additional insights with respect to Bonnaire et al. (2025), that are indeed there.

W3. Some images can be made more clear and informative for the sake of presentation (see further parts of the review).

**Audience:**

Yes

**Audience Explanation:**

In my personal opinion, the paper fully fits the interests of TMLR's readers audience, both the more theory-oriented ones and practitioners. The first part of the audience will find in the paper a derivation, an explanation, and a subsequent validation, of the memorization dynamics in Diffusion Models, that finally sheds light to the difficulty of such models of retrieving training data. The second part of the readership will instead gain indications about optimal early-stopping times and useful insights about the training of diffusion models from text and risk of partial memorization.

**Broader Impact Concerns:**

No concern on any ethical implication of this work.

**Claims And Evidence:**

Yes

**Claims Explanation:**

The claims made in the paper, as well as the theoretical argument provided, are supported by a rich experimental work across several dataset structures (i.e. images, text, synthetic grammar model), parametric models (e.g. iDDPM, Stable Diffusion v2.1, one-hidden-layer fully-connected network in the NTK regime) and training algorithms. All experiments, as well as further validation from concurrent works in the literature, do support the generality of the claim about the separation between generalization and memorization learning phases in diffusion models.

**Requested Changes:**

I will now enumerate a short list of adjustments aimed at improving the quality of the manuscript:

C1. Fig.1 (right) is not very informative: I suggest to show progression of generated images at different training times, highlighting the ones obtained at $\tau_{mem}$.

C2. I believe that Authors would improve fig.2 by including $\tau = \tau_{mem}$ in the right panel. Plus a question: do the curve start from high values at small times in figures 2 (blue one) and 5(b) ?

C3. In text diffusion models, is dim(x) = context size ? This is not very clear from the text.

C4. In Section 3.1 Authors quantify the quality of generalization around $t_{gen}$ measuring FID (e.g. Section 3.1). This quantity is supposed to be small in two different instances: the model samples realistic images; the model retrieves training images. I believe that the Authors should also report FID inside memorization as a baseline, and compare with FID at generalization.

C5. As stressed in W2, I would suggest the Authors to be even more straight and clear about the interpretation of the O(P) time scale for memorization, in Section 4. I would explain that the reason is not related to the progressive evaluation of the single data-points, as someone more familiar with supervised learning would think, but it is rather proportional to the number of spikes in the empirical data distribution at small sampling times, and the way they are related to slow-learning modes of the kernel. I would also use this argument as an additional comparison with Bonnaire et al. (2025) who, instead, do not provide for an interpretation, but only rely on the structure of the feature-feature correlations in that particular parametric model.

---

> ### Author Response · Authors · 2026-05-12
>
> We thank the reviewer for the careful and constructive comments. In the revised manuscript -- which we will upload in the coming days --, we will upgrade Fig. 1 (showing FID and a progression of generations through training, clarify the meaning of $\dim(x)$, and add a paragraph in Sec. 4 that gives the phenomenological reading of the $O(P)$ memorization timescale.
>
> **Figure 1 (right) (C1)**
> The revised figure will combine two of the changes requested across reviews: display the FID curve next to the losses and the copy fraction, and show generations at several training times.
>
> **Behaviour at small training times in Figures 2 and 5(b) (C2)**
> At very small $\tau$ both networks produce outputs that match because they are both approximating the same population score, and only at $\tau_{\mathrm{mem}}$ the two outputs separate, each collapsing toward its own training set.
>
> **$\dim(x)$ in language experiments (C3)**
> $\dim(x)$ is the context length in characters (256 in our setup). We have made this explicit in Appendix B (Sec. B.2).
>
> **FID inside the memorization regime (C4)**
> The reviewer's intuition that the FID stays low during memorization implicitly assumes the FID is computed against the same set that the model was trained on, in which case copying the training set would indeed collapse the FID. We instead evaluate FID against a reference distinct from the training subset the model saw. For CIFAR-10, the reference is the union of the standard train and test splits, which is much larger than the training subset $P\le 16{,}384$ used by the model, so a model that collapses onto its training examples loses diversity relative to the reference and is penalized. The FID dynamics in Appendix Fig. 10 (now incorporated into Fig. 1) bear this out: the FID decreases as the model learns the population distribution, reaches a minimum at $\tau\approx\tau_{\mathrm{mem}}$, and then *grows* as the model enters memorization and loses diversity. The FID measured deep in the memorization phase is therefore substantially worse than the FID at $\tau_{\mathrm{mem}}$, and the early-stopping window is bounded from both sides by an FID criterion. We will add one sentence to this effect in Sec. 3.
>
> **Phenomenological interpretation of the $O(P)$ scaling (C5)**
> The key intuition we will make explicit is the following. A reader more familiar with supervised learning might expect the $O(P)$ scaling to come from the optimizer: at fixed batch size, each training point is revisited a number of times proportional to $P$ over a fixed number of training steps, so one would naively predict the time to fit the data to scale with the number of passes through the dataset. This is *not* what is happening, and we rule it out directly by varying the batch size from $B=8$ to full-batch $B=P$ in Fig. 18, where $\tau_{\mathrm{mem}}$ is unchanged. The correct picture is that the $P$ in $\tau_{\mathrm{mem}}\propto P/\sigma^{\nu}$ is a property of the empirical data distribution itself. At low diffusion noise $\sigma$, the empirical score $g_{\sigma}$ decomposes into $P$ localized, sample-specific spikes, each centered on a training point and supported on a ball of radius $O(\sigma)$. In the empirical density $p_{\sigma}$ each such spike carries mass $1/P$. Fitting these spikes requires exciting localized modes of the kernel integral operator $T_K$ on $L^2(p_{\sigma})$, and the $1/P$ mass per spike pushes the corresponding eigenvalues down to the spectral scale $\sigma^{\nu}/P$ (Proposition 1). The inverse of this scale is the time required to learn these modes, hence $\tau_{\mathrm{mem}}\propto P/\sigma^{\nu}$. The $P$ in the numerator is therefore the number of $1/P$-weighted localized spikes that the model must fit in order to memorize, *not* the number of times the optimizer revisits each individual point.
>
> We hope these revisions address the reviewer's points and would be glad to refine any of the additions further.

---

### Review · Reviewer_8nDh · 2026-04-28

**Summary Of Contributions:**

This paper studies how overparameterized neural networks manage to generalize during the training dynamics before eventually memorizing the training set. Crucially, the authors establish with numerical experiments as well as analytical arguments that the typical timescale to memorize $\tau_{\text{mem}}$ scales linearly with the number of samples $P$ in the training set. Thus there exists a large window where one can early stop the training dynamics and obtain good generalization properties without memorization. It is an interesting scientific problem since the mechanisms that enable diffusion models to generalize (i.e. generate new samples from a target distribution without reproducing the training set) are not fully understood.

**Strengths**

- The paper addresses an interesting and timely scientific question

- The authors conducted large scale numerical experiments on a variety of data distributions for vision diffusion models (iDDPM on CIFAR-10 and CelebA, Stable Diffusion fine-tuned on LAION-10k) as well as for discrete diffusion with Language Diffusion Models, which is rare in theoretical papers on this subject. It is in my opinion the main strength of the paper.

- The authors present a theoretical understanding of this phenomenon with kernel regression.

- They extend the experiments of Kadkhodaie et al. (2023) on the overlap between two models trained on distinct datasets. The authors show that before $\tau_{\text{mem}}$, the two models generate the same samples while after $\tau_{\text{mem}}$ they generate samples from their respective training sets.

- The authors show that the scaling $\tau_{\text{mem}} \propto P$ is robust to the batch size and the choice of optimizer.

- They study the Random Hierarchy Model (RHM) as a toy model where they show a competition between the different timescales for partial generalization and the timescale to memorize and they provide a phase diagram.

- The differences between the contemporaneous work Bonnaire et al. (2025) are discussed, as well as the difference between overfitting in diffusion models and in supervised learning.

**Weaknesses**

- For vision Diffusion Models, the figures in the main text only show the evolution of the train/test losses and the fraction of copied images but it would also be relevent to put along side the evolution of the FID. I believe that the only figure showing the evolution of FID vs training time in Figure 10 in the appendix.

- Fig 2: Rather than a transition from generalization to memorization there seems to be a transition from bad generalization/ non-memorization to memorization since the samples obtained for $\tau<\tau_{\text{mem}}$ are of worse quality than the images in the training set.

- In the theoretical part, the authors only estimate the memorization timescale $\tau_{\text{mem}}$. However, the claim of a 'large window' for early stopping requires a characterization of the generalization timescale ($\tau_{\text{gen}}$) and showing that the ratio $\tau_{\text{mem}}/\tau_{\text{gen}}$ scales linearly with $P$. I would be interested to know the difference between the eigenfunctions that 'generalize' and the ones that memorize for instance regarding their localization properties.

**Audience:**

Yes

**Audience Explanation:**

Understanding how generative models learn from a finite dataset to create novel samples rather than merely reproducing training data is a fundamental open question in machine learning. This paper is of high interest to the TMLR audience because it provides an explanation of how memorization is prevented in overparameterized regimes for both vision and language diffusion models. The establishment of the linear scaling law for the memorization timescale ($\tau_{\text{mem}} \propto P$) is a valuable theoretical result that challenges common assumptions about overparameterization and benign overfitting. Furthermore, this finding has direct practical value for practitioners, as it provides a framework for hyperparameter scaling.
Despite the existence of contemporaneous work on similar subjects such as Bonnaire et al. (2025), this paper provides a distinct contribution (a large variety of data distributions, a different analytical model, etc.) that would be of interest to the community.

**Broader Impact Concerns:**

No concern on any ethical implication of this work.

**Claims And Evidence:**

Yes

**Claims Explanation:**

The authors support their results with large-scale numerical experiments across several modalities, including vision (CIFAR10, CelebA, LAION) and language (MD4), using large training sets (from 2,048 to 16,384 for images and 16,000 to 262,000 tokens for text data). The models studied are large-scale neural networks (0.5 billion parameters for the U-Net used for images and 165M for the GPT-like transformer for text), demonstrating that the described phenomenon is robust across different architectures. The authors also study the Random Hierarchy Model as a toy model to characterize the competition between the timescales for generalization and memorization. Regarding the theoretical claims, they examine the case of kernel regression where the timescales of the training dynamics are linked to the inverse eigenvalues of the kernel. The authors use heuristic arguments to identify the scaling of the eigenvalues associated with memorization and back these claims with simulations of two-layer neural networks in the lazy regime.

**Requested Changes:**

A change that could strengthen the paper on the theoretical side would be to perform an analysis of the 'generalization' eigenfunctions/values. Can they be characterized? Is the scaling of the eigenvalues $O_P(1)$? Are the eigenfunctions localized/delocalized? However, it might be out of the scope of the paper, which is mainly experimental, so I do not mind if the paper is accepted without this addition.

---

> ### Author Response · Authors · 2026-05-12
>
> We thank the reviewer for the careful reading and for highlighting the breadth of our experiments and the value of our theoretical argument. In the revised manuscript -- which we will upload in the coming days -- we will move the FID curve into Fig. 1 so that it sits next to the loss and copy-rate dynamics, expand the discussion of Fig. 2 to make explicit that it documents *progressive* generalization (a phenomenon distinct from full generalization), and add a paragraph in Sec. 3 characterizing the generalization timescale $\tau_{\mathrm{gen}}$ relative to $\tau_{\mathrm{mem}}$. Detailed responses follow in the order of the review.
>
> **FID in the main text (W1)**
> We agree and will move the FID dynamics (previously Fig. 10) into the main-text Fig. 1.
>
> **Figure 2 (W2)**
> The intent of this experiment is, in fact, to demonstrate *progressive* generalization, which we believe is a new finding and which extends the static notion of strong generalization of Kadkhodaie et al. The two models in Fig. 2 are trained on disjoint subsets and yet, throughout the entire pre-memorization window, generate visually almost identical images from the same noise seed. This is the key point: the function being learned during $\tau<\tau_{\mathrm{mem}}$ is a property of the *population* distribution, not of the empirical one, and is therefore the same for both models. The fact that the shared images are not yet of training-set quality is a consequence of running this experiment at a relatively small $P=2{,}048$: with this $P$ the model has not yet reached full generalization when memorization sets in. This is exactly the partial-generalization regime that we characterize quantitatively in the RHM (Sec. 5, Fig. 5(a)), where the model first acquires shallow rules common to the population and only later, under memorization pressure, fits the remaining sample-specific structure.
>
> **Generalization timescale and "generalization eigenfunctions" (W3)**
> Across all our experimental settings, $\tau_{\mathrm{gen}}$ is a property of the *target distribution*, not of the training set size $P$. This is visible directly in the data: when the loss curves of Figures 1, 3 and 4 are rescaled as $\tau/P$ (insets), the validation curves at different $P$ collapse onto a common decay phase but *begin* to decrease at a $P$-independent absolute training time. This is exactly the statement that $\tau_{\mathrm{gen}}=O_{P}(1)$, and combined with $\tau_{\mathrm{mem}}\propto P$ it gives $\tau_{\mathrm{mem}}/\tau_{\mathrm{gen}}\propto P$. The RHM provides a fully controlled instance of this dichotomy: layer-$\ell$ generalization corresponds to learning compositional features that span entire subtrees, with sample complexity $P*_{\ell}$ that depends on the grammar but not on the available $P$ once $P>P*_{\ell}$; the corresponding timescale is therefore $P$-independent, while the memorization scale grows linearly with $P$.
>
> At the spectral level, the picture is consistent: the localized odd probes $\phi_{i,u}$ constructed in Appendix G sit at eigenvalue scale $\sigma^{\nu}/P$ (Proposition 1), are explicitly supported on a ball of radius $O(\sigma)$ around a single training point, and shrink as $P$ grows; these are the "memorization" modes. The complementary "generalization" modes are functions that probe the population density rather than individual samples. For those modes, the empirical measure can be replaced by the population measure with corrections $O(1/\sqrt{P})$, giving eigenvalues that are $O_{P}(1)$ (see argument below).
>
> We hope these revisions address the reviewer's concerns and welcome any further questions.

---

> > ### Author Response · Authors · 2026-05-12
> >
> > ### Generalization timescale in the kernel regime
> >
> > Define the *population* kernel operator $\bar T_K f(x) = \int K(x,y)f(y)\bar p_\sigma(y)\,dy$ on the smoothed population $\bar p_\sigma$, and let $g = \nabla \log \bar p_\sigma$ be the population score. Expand $g$ in the eigenbasis of $\bar T_K$ ($T_K \varphi_n = \lambda_n \varphi_n$): $g = \sum_n a_n\,\varphi_n$ and define the generalization eigenvalue
> > $$
> > \lambda_{\mathrm{gen}} := \min\{\lambda_n : a_n \neq 0\},
> > $$
> > the slowest spectral mode the population score excites. The eigenfunctions $\{\varphi_n\}$, eigenvalues $\{\lambda_n\}$, and coefficients $\{a_n\}$ depend on the kernel, the noise level, and the population distribution but not on the training sample. In particular $\lambda_{\mathrm{gen}}$ is independent of $P$. This extends from $\bar T_K$ to the *empirical* operator $\hat T_K$ by concentration: $\hat p_\sigma$ is a sample average of $P$ smooth Gaussian bumps and concentrates around $\bar p_\sigma$ at the classical $P^{-1/2}$ rate. The matrix elements of $\hat T_K$ converge to those of $\bar T_K$ at the same rate. The eigenvalues of $\hat T_K$ on the population modes therefore remain $O_P(1)$ as $P\to\infty$. Thus, under gradient flow,
> > $$
> > \tau_{\mathrm{gen}} \sim 1/\lambda_{\mathrm{gen}} = O_P(1),
> > \qquad
> > \tau_{\mathrm{mem}} \sim P.
> > $$
> > The ratio $\tau_{\mathrm{mem}}/\tau_{\mathrm{gen}} \propto P$ is exactly the growing window the paper identifies.
> >
> > When $g$ has spectral weight at arbitrarily high modes, $\lambda_{\mathrm{gen}}$ as defined above can vanish. In such cases, we can define
> > $$
> > \lambda_{\mathrm{gen}}(\epsilon)
> > :=
> > \sup \lbrace \lambda:\sum_{\lambda_n < \lambda} a_n^2 \leq \epsilon \rbrace ,
> > $$
> > i.e., the spectral cutoff below which the population score's mass does not exceed $\epsilon$. Generalization to error $\epsilon$ is reached at $\tau_{\mathrm{gen}}(\epsilon) \sim 1/\lambda_{\mathrm{gen}}(\epsilon)$, still intensive in $P$. For a power-law spectrum $\lambda_n \sim n^{-\beta}$, and $a_n^2 \sim n^{-\alpha}$ with $\alpha > 1$, the tail condition $\sum_{n>N} a_n^2 \leq \epsilon$ gives $N \sim \epsilon^{-1/(\alpha-1)}$, hence $\lambda_{\mathrm{gen}}(\epsilon) \sim \epsilon^{\beta/(\alpha-1)}$ and $\tau_{\mathrm{gen}}(\epsilon) \sim \epsilon^{-\beta/(\alpha-1)}$, independent of $P$. The two-timescale picture survives whenever $\tau_{\mathrm{gen}}(\epsilon) \ll \tau_{\mathrm{mem}} \sim P$, which holds at any fixed $\epsilon$ for sufficiently large $P$.

---

### Author Response · Authors · 2026-05-20
**Revision posted**

Dear reviewers,

We have uploaded a revised manuscript incorporating the changes promised in our responses. Main modifications are highlighted in red.

Best,

The authors

---

### Decision · Action_Editor_Kso5 · 2026-06-08

**Recommendation:** Accept as is

**Audience:**

Yes

**Audience Explanation:**

The theory of diffusion models is a topic of interest to the TMLR community, and this paper provide interesting contributions on how generalisation and memorisation in diffusion can be understood in terms of an interplay of timescales driven by overparametrisation.

**Claims And Evidence:**

Yes

**Claims Explanation:**

The paper studies the training dynamics of over-parameterised diffusion models and shows, across image, language, and synthetic hierarchical data, that models can generalise before eventually memorising, with the onset of memorisation scaling approximately linearly with the training set size. The empirical findings are complemented by a kernel-based scaling argument and a controlled Random Hierarchy Model phase diagram supporting early stopping as a principled way to avoid memorisation.

The main concerns concerned the clarity and scope of the theoretical explanation, its positioning with respect to the contemporaneous work of Bonnaire et al., and some presentation issues in the empirical figures and metrics. Following the discussion, the authors substantially clarified the theoretical framework and its assumptions, added discussion of the relevant generalisation modes and the interpretation of the memorisation timescale, improved the figures and presentation, and more clearly articulated the relation to prior and concurrent work.

All reviewers recommend acceptance, and I agree with their assessment. The paper addresses a timely and important question in generative modelling, namely why overparameterised diffusion models can produce novel samples before entering a memorisation regime. The submission provides extensive and convincing empirical evidence across multiple modalities and architectures, together with a useful theoretical interpretation of the observed linear scaling of memorisation time with dataset size. The concerns raised during review were adequately addressed in the author discussion and revisions. I therefore recommend acceptance.